# An Efficient BDS-3 Long-Range Undifferenced Network RTK Positioning Algorithm

**Huizhong Zhu *** , **Jie Zhang, Jun Li and Aigong Xu**

School of Geomatics, Liaoning Technical University, Fuxin 123000, China; 472120783@stu.lntu.edu.cn (J.Z.); 472010048@stu.lntu.edu.cn (J.L.); xuaigong@lntu.edu.cn (A.X.)
* Correspondence: zhuhuizhong@lntu.edu.cn; Tel.: +86-010-6811-1077

**Abstract:** In 2020, the BeiDou-3 global navigation satellite system (BDS-3) was officially completed and put into service. Currently, network real-time kinematic (RTK) technology is considered the main means through which to improve the positioning accuracy of the BeiDou navigation satellite system (BDS). This paper proposes a long-range undifferenced network RTK (URTK) algorithm, based on multi-frequency observation data of the BDS. First, the multi-frequency phase integer ambiguity resolution (AR) model considering atmospheric error parameters is designed, and the multi-frequency phase integer ambiguity of the long-range BDS reference station is determined. Then, the undifferenced integer ambiguity of each reference station is obtained, using linear variation based on the accurately determined phase integer ambiguity between reference stations, and the undifferenced observation error of each reference station is calculated. Considering the weakening spatial correlation of the observation errors between long-range stations, undifferenced classification error corrections of a reference station network are separated, according to different error characteristics. Finally, the inverse distance weighting method is employed to calculate the classification undifferenced error correction of the rover station. The rover station corrects the observation error through applying the undifferenced error correction to achieve high-precision positioning. The measured data of a long-range continuous operation reference station (CORS) network are selected for an experiment. The results show that the proposed algorithm can quickly and accurately realize the resolution of the BDS integer ambiguity of a reference station network and establish an undifferenced area error correction model in order to achieve accurate classification of undifferenced error correction values for a rover station. In China, the BDS-3 is superior to the global positioning system (GPS) in terms of the satellite number, position dilution of precision (PDOP) value, AR success rate, stability, and convergence time. The results show that the AR success rate, stability, and convergence time increase with the operational frequency, and the BDS-3 can achieve centimeter-level positioning of single-system rover stations without relying on the GPS.

**Keywords:** BDS-3; multi-frequency data; long-range; network RTK; classification error; undifferenced error correction

## 1. Introduction

On 31 July 2020, the BeiDou-3 global navigation satellite system (BDS-3), which was independently developed by China, was officially completed and entered its application stage, providing the positioning, navigation, and timing (PNT) services for global users, and representing the successful completion of the BDS construction process [1]. The BDS-3 provides various services, including international search and rescue, short message communication, and satellite-based enhancement, which significantly enhances its application scope and global influence. Compared to the BeiDou-2 navigation satellite system (BDS-2), the BDS-3 has significantly improved coverage, spatial signal accuracy, spatial signal availability, and spatial signal continuity [2]. The BDS-3 constellation consists of 24 medium Earth orbit (MEO), three geostationary Earth orbit (GEO), and three inclined geostationary

orbit (IGSO) satellites. The BDS-3 and BDS-2 have two of the same frequencies, B1I and B3I, but the B1C, B2a, and B2b frequencies have been added to the BDS-3 for the broadcast of B1I (1561.098 MHz), B2b (1207.14 MHz), B3I (1268.52 MHz), B1C (1575.42 MHz), and B2a (1176.45 MHz) signals. The increase in observation frequency has addressed data redundancy and improved positioning performance. Moreover, additional frequencies have provided a guarantee for the ambiguity resolution (AR) of the carrier phase in the solution process, and the positioning accuracy can also be significantly increased. Particularly under poor observation conditions, more frequencies can provide better positioning results. Furthermore, a larger number of frequencies improves compatibility and interoperability with other global satellite navigation systems (GNSS). The B1I and B3I signals of the BDS-3 are compatible with those of the BDS-2, and the B1C and B2a signals are compatible with the global position system (GPS) L1/L5 and Galileo E1/E5a signals, which provides favorable conditions for combining the BDS-3 with other systems. In [2], the authors studied the PNT, satellite-based augmentation, precise point positioning, short message communication, and Cospas-Sarsat performances of the BDS-3. In [3], the signal-to-noise ratio, pseudo-range observation error, and multipath error of the BDS-3 test system were studied and compared with those of the BDS-2. In [4,5], the overall design, coordinate reference system, time reference system, and the basic performance of the BDS-3 were introduced. The aforementioned studies provide an important reference base for the study of the BDS-3.

The distance between reference stations is generally tens or hundreds of kilometers. The correlation between the atmospheric delay and the satellite orbit error is low, so the residual error of observation equations is larger than half the wavelength of the phase observation. Even when a reference station's coordinates are known, it is difficult to separate the integer ambiguity from the error. Therefore, the issue of how to address the integer ambiguity of the reference station has been one of the main problems in the undifferenced network real-time kinematic (URTK) algorithms [6]. The URTK algorithms include the development of a regional error model and the calculation of the user phase integer ambiguity. The network real-time kinematic (RTK) has been widely used in many fields, including deformation monitoring, cadastral surveying, and automatic driving [7–10]. In [11], the network RTK was used to correct errors in local areas in order to improve the positioning accuracy of the RTK. In recent years, extensive research has been conducted on the AR of the network RTK reference stations. In [12,13], the ionospheric model constraint method was used to improve a reference station's AR speed. In [14], an improved AR method for long-range reference stations with double tropospheric parameter restrictions was proposed. In [15], the authors proposed an ionospheric-free (IF) three-carrier ambiguity resolution (TCAR) method through which to resolve the ambiguity problem between the reference stations over long baselines. Further, in [16], the correlation between the baselines, composed of reference stations, was considered, and an efficient method of determining the AR of reference stations was developed based on the network solution mode to improve the accuracy of float ambiguity and accelerate the convergence speed of ambiguity. In [17], the optimal ambiguity subset was obtained via the optimized partial ambiguity solution method, which improved the AR rate between reference stations. In [18], a long-range BDS triple-frequency phase integer AR method, which considered the actual atmospheric delay variation constraint and integer ambiguity constraint of GEO satellites, was developed. In [19], a single-epoch determination method of triple-frequency phase integer ambiguity for a BDS reference station was studied. Recently, there have been many studies on network RTK algorithms. In [20], an undifferenced algorithm was proposed for network RTK; this algorithm uses the regional undifferenced error correction to correct the error of a rover station and realize its positioning. In [21], the authors conducted a comparative analysis of the double-difference model and the undifferenced model of the network RTK using the observed data of the BDS-2. A long-range network RTK method for the BeiDou satellite navigation system was studied in [22]. Currently, the GPS is the most mature and widely used GNSS. In [23–25], the network RTK positioning algorithm for the hybrid BDS-2–GPS system was studied under atmospheric constraints. In addition, much research on the

BDS-3 has been conducted in recent years. In [26,27], the authors mainly studied the network RTK performance of the BDS-3 at the B1C and B2a frequencies, and the results showed that the overall performance of B1C/B2a was better than that of B1I/B3I. In [28], the long-range network RTK of the BDS-3 four-frequency ionospheric weighted model was studied in order to improve its AR rate. The aforementioned studies mainly considered the BDS-2 and GPS systems, along with the performance of the BDS-3. However, there have been fewer studies on the long-range URTK algorithm of the BDS-3.

This paper studies a BDS-3 single-system multi-frequency long-range undifferenced network RTK method. The proposed method makes full use of the advantages of the BDS-3 full constellation to provide multi-frequency observation data. First, the phase integer ambiguity of a reference station network is calculated using the random walk constraint model of the atmospheric delay error between epochs. Then, the classification undifferenced error correction of the reference station network is realized, and the classification undifferenced error correction of the rover station is obtained via interpolation, using an inverse distance weighting algorithm. The phase integer AR and position parameter calculation of the multi-frequency BDS-3 data is performed. Finally, the BDS-3 long-range URTK positioning algorithm is verified and analyzed experimentally using the measured continuous operation reference station (CORS) network data.

## 2. Long-Range URTK Algorithm

The long-range URTK algorithm is mainly composed of three modules, denoted as Module 1, Module 2, and Module 3, which are explained in detail in the following paragraphs.

Module 1: Using an integer AR model that considers the atmospheric error, the baseline in the GNSS reference station network is solved in order to determine the double-difference integer ambiguity. If the double-difference ambiguity cannot be fixed, the next epoch is directly solved.

Module 2: Then, the double-difference integer ambiguity is converted into the undifferenced integer ambiguity. Via substituting the undifferenced integer ambiguity of the reference station network into the observation equation, the undifferenced error correction of each baseline is realized, and the undifferenced error correction of the rover station is interpolated using the inverse distance weighting algorithm.

Module 3: The classification undifferenced error correction data of the rover station are used in the observation equation of the rover station in order to calculate the position parameters, and the real-time high-precision positioning of a user is achieved. The flowchart of the long-range URTK is shown in Figure 1.

As shown in Figure 1, Module 1 has the most complex structure among the three modules, and it includes two main parts. The first part indicates that, when constructing the pseudo-range and phase double-difference observation equations, the atmospheric delay needs to be constrained by an inter-epoch random walk model. However, this constraint is indispensable for long-range network RTK, which directly affects the accuracy of the floating solution of double-difference ambiguity, thus influencing the convergence speed of double-difference ambiguity. The second part relates the test of double-difference ambiguity between reference stations, which can ensure the accuracy of double-difference AR. Module 2 relates mainly to the regional error model construction, which is performed on the premise that Module 1 can obtain an accurate double-difference AR. In Module 2, the crucial tasks are the calculation of undifferenced classification error correction and the interpolation of undifferenced error correction for a rover station. The result of this task directly affects the final positioning result of Module 3.

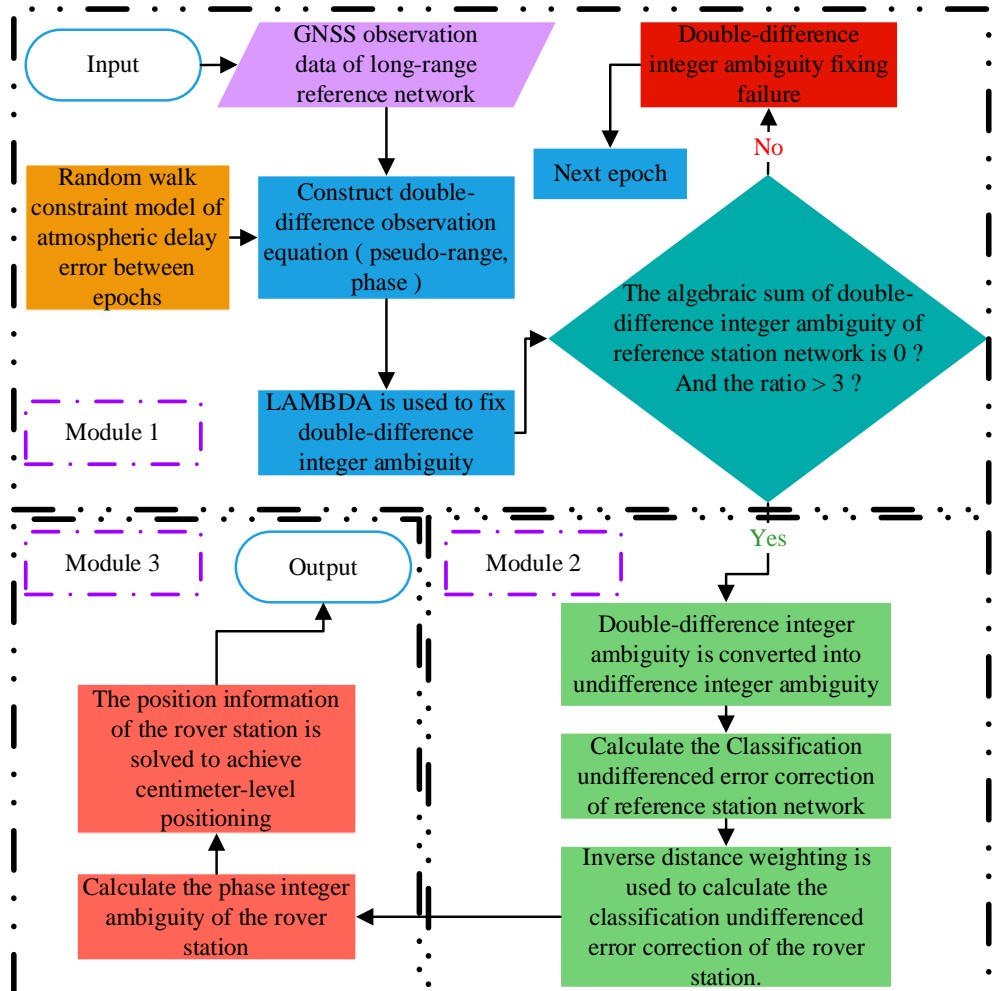

**Figure 1.** The flowchart of the long-range URTK algorithm.

## 2.1. Reference Station Network Double-Difference Integer AR

Due to the long range, of more than 100 km, between the long-range network RTK reference stations, the effect of ionospheric and tropospheric delays on the double-difference observations is much stronger than that of the 0.5 cycle. Even when the multi-frequency observation data and the reference station coordinates are known, the integer ambiguity and error are difficult to separate. In this study, the undifferenced observation equations of phase and pseudo-range linearization used in the long-range network RTK algorithm are respectively expressed as follows [22]:

$$\lambda_i \varphi_i = H \cdot X + \rho_r^s + c(t_r - t^s) + O^s - I_{r,i}^s + T^s - \lambda_i(N_{r,i}^s + \sigma_r - \sigma^s) + \varepsilon_{r,i}^s \tag{1}$$

$$P_i = H \cdot X + \rho_r^s + c(t_r - t^s + d_r - d^s) + O^s + I_{r,i}^s + T^s + \delta_{r,i}^s \tag{2}$$

where $\lambda_i$ is the phase observation wavelength; $\varphi_i$ denotes the phase observation data; $H$ is the coefficient of a position parameter $X$; $\rho_r^s$ is the distance between stations and satellites; $c$ is the speed of light in a vacuum; $t_r$ is the receiver clock error; $t^s$ is the satellite clock error; $O^s$ is the satellite orbit error; $I$ denotes the ionospheric delay error; $T$ represents the tropospheric delay error; $N$ is the integer ambiguity; $\sigma_r$ is the phase hardware delay of a receiver; $\sigma^s$ is the phase hardware delay of a satellite; $\varepsilon_{r,i}^s$ is the phase observation noise; superscript $s$ represents the satellite number, and subscript $i$ represents different frequencies; $P$ is the pseudo-range observation value; $d_r$ is the pseudo-range hardware delay of the

receiver; $d^s$ is the pseudo-range hardware delay of the satellite; and $\delta_{r,i}^s$ represents the pseudo-range measurement noise.

Consider the synchronous observation of the first-frequency carrier phase observations of a reference satellite $q$ and a satellite $p$, obtained via the three reference stations denoted as $A$, $B$, and $C$, where the coordinates of the reference station are accurately known; then, the observation equation for the carrier phase integer AR of the reference station is obtained as follows:

$$
\begin{cases}
\Delta\nabla L_{AB}^{pq} = (I_{A,i}^p - I_{B,i}^p) - (I_{A,i}^q - I_{B,i}^q) + (Map_A^p - Map_A^q)RZTD + \lambda_i\Delta\nabla N_{AB}^{pq} \\
\Delta\nabla L_{BC}^{pq} = (I_{B,i}^p - I_{C,i}^p) - (I_{B,i}^q - I_{C,i}^q) + (Map_B^p - Map_B^q)RZTD + \lambda_i\Delta\nabla N_{BC}^{pq} \\
\Delta\nabla L_{CA}^{pq} = (I_{C,i}^p - I_{A,i}^p) - (I_{C,i}^q - I_{A,i}^q) + (Map_C^p - Map_C^q)RZTD + \lambda_i\Delta\nabla N_{CA}^{pq}
\end{cases}
\tag{3}
$$

where,

$$
\begin{cases}
\Delta\nabla L_{AB}^{pq} = (\rho_B^p - \rho_B^q - \rho_A^p + \rho_A^q) - (\varphi_B^p\lambda_i - \varphi_B^q\lambda_i - \varphi_A^p\lambda_i + \varphi_A^q\lambda_i) \\
\Delta\nabla N_{AB}^{pq} = (N_{B,i}^p - N_{B,i}^p) - (N_{A,i}^q - N_{A,i}^q)
\end{cases}
\tag{4}
$$

where $\Delta\nabla$ is a double-difference operator, $L$ is a constant term, $Map$ is the tropospheric projection function, and $RZTD$ denotes the relative zenith tropospheric delay error of two reference stations.

The main errors in the observation equation are non-dispersive errors, which are dominated by tropospheric delay and dispersive errors [12]. The GMF projection function and the zenith tropospheric delay error are used to represent the tropospheric delay error in a reference station's observations [29]. The projection function values of the same satellite of two adjacent reference stations are close, and the zenith tropospheric delay error is included in the relative zenith tropospheric delay error parameter. Although the double-difference observation equation can eliminate most of the errors (e.g., satellite and receiver clock errors, orbital errors, hardware delays at the satellite and receiver, and observation noise), for long-range RTK, if reasonable measures are not used to process the atmospheric delay, the fixed speed and correctness of the double-difference ambiguity will be significantly affected. Therefore, in order to prevent the rank defect problem caused by the estimation of atmospheric delay parameters in each epoch, and to consider the time-varying property of atmospheric delay, this study processes the ionospheric and tropospheric parameters using a random walk model-based method, considering the constraint between epochs. The random walk process used for the troposphere and ionosphere is defined as follows [30,31]:

$$
\begin{cases}
RZTD_t - RZTD_{t+1} = \omega_T, \omega_T \sim N(0, \sigma_T^2) \\
I_t - I_{t+1} = \omega_I, \omega_I \sim N(0, \sigma_I^2)
\end{cases}
\tag{5}
$$

where $RZTD_t$ and $RZTD_{t+1}$ are the relative zenith tropospheric delays of epochs $t$ and $(t+1)$, respectively; $I_t$ and $I_{t+1}$ are the ionospheric delays of the two epochs $t$ and $(t+1)$, respectively; $\omega_T$ and $\omega_I$ represent the zenith tropospheric and ionospheric parameters of the epoch difference, respectively.

Equation (5) indicates that the atmospheric parameter variations at different observation times can be expressed with a random walk process. The zenith tropospheric parameters satisfy a normal distribution, with a mean of zero and a variance of $\sigma_T^2$. In addition, the ionospheric parameters also satisfy a normal distribution, with a mean value of zero and a variance of $\sigma_I^2$. As long as an appropriate variance can be obtained, the corresponding constraints on atmospheric parameters can be imposed. In this study, the tropospheric delay power spectral density is 1 cm/$\sqrt{h}$, and the ionospheric delay power spectral density is 1 m/$\sqrt{h}$ [32]. After the power spectral density is given, the corresponding variance can be calculated, and the weight of the constraint equation can be calculated

using the variance. The specific expression of the virtual observation equation of the atmospheric parameter random walk process is as follows:

$$
\begin{cases}
v_T = RZTD_t - RZTD_{t+1}, \beta_T = \dfrac{\sigma_0^2}{\sigma_T^2} = \dfrac{\sigma_0^2}{\phi_T^2 \cdot \Delta t} \\
\quad v_I = I_t - I_{t+1}, \beta_I = \dfrac{\sigma_0^2}{\sigma_I^2} = \dfrac{\sigma_0^2}{\phi_I^2 \cdot \Delta t}
\end{cases}
\tag{6}
$$

where $\beta_T$ and $\beta_I$ represent the weights of the tropospheric delay and ionospheric delay virtual observation equations, respectively; $\phi_T$ and $\phi_I$ represent the power spectral density of tropospheric delay and ionospheric delay, respectively; and $\Delta t$ represents the time interval of atmospheric variations. The relationship between variance and power spectral density can also be seen directly from Equation (6).

Assuming that there are ($s + 1$) common-view satellites between reference stations $A$ and $B$, the normal equation of the BDS-3 multi-frequency observation data is as follows:

$$
\begin{bmatrix} V_1^S \\ \vdots \\ V_n^S \end{bmatrix} = \begin{bmatrix} Map^S & Iono^S & \lambda^S \end{bmatrix} \begin{bmatrix} RZTD_{t,t+1} \\ I_{t,t+1}^S \\ N^S \end{bmatrix} - \begin{bmatrix} L_1^S \\ \vdots \\ L_n^S \end{bmatrix}
\tag{7}
$$

where,

$$
V_i^S = \begin{bmatrix} V_i^{S_{1,2}} \\ \vdots \\ V_i^{S_{1,s+1}} \end{bmatrix}_{s \times 1} \quad
Map^S = \begin{bmatrix} Map_1^{S_{1,2}} & 0 \\ \vdots & \vdots \\ Map_n^{S_{1,s+1}} & 0 \end{bmatrix}_{(n \times s) \times 2} \quad
Iono^S = \begin{bmatrix} b & 0 \\ (\frac{f_1^2}{f_2^2})b & 0 \\ \vdots & \vdots \\ (\frac{f_1^2}{f_n^2})b & 0 \end{bmatrix}_{(n \times s) \times [2 \times (s+1)]} \quad
\lambda^S = \begin{bmatrix} \lambda_1^S b & \cdots & 0 \\ \vdots & \ddots & \vdots \\ 0 & \cdots & \lambda_n^S b \end{bmatrix}_{(n \times s) \times [n \times (s+1)]}
$$

$$
RZTD_{t,t+1} = \begin{bmatrix} RZTD_t \\ RZTD_{t+1} \end{bmatrix}_{2 \times 1} \quad
I_{t,t+1}^S = \begin{bmatrix} I_t^S \\ I_{t+1}^S \end{bmatrix}_{[2 \times (s+1)] \times 1} \quad
N^S = \begin{bmatrix} \Delta N_1^{S_1} \\ \vdots \\ \Delta N_n^{S_{s+1}} \end{bmatrix}_{[n \times (s+1)] \times 1} \quad
L_i^S = \begin{bmatrix} L_i^{S_{1,2}} \\ \vdots \\ L_i^{S_{1,s+1}} \end{bmatrix}_{s \times 1} \quad
b = \begin{bmatrix} 1 & 0 & -1 & 0 & \cdots & 0 \\ 0 & 1 & -1 & 0 & \cdots & 0 \\ 0 & 0 & -1 & 1 & \ddots & 0 \\ \vdots & \ddots & \ddots & \ddots & \ddots & \vdots \\ 0 & 0 & -1 & 0 & 0 & 1 \end{bmatrix}_{(s) \times (s+1)}
$$

where superscript $S$ represents different systems; subscript $i$ represents different frequencies; $n$ denotes the maximum number of frequencies; $f_i$ represents the corresponding frequency of observations; and $\Delta N$ is the single-difference integer ambiguity vector between stations.

The atmospheric epoch constraint is introduced and defined according to the random walk process of the ionosphere and troposphere. The mean value of the double-difference between the ionosphere and the troposphere in the two epochs is set to zero, as the constraint condition between epochs. Further, in order to enhance the properties of Equation (7) and improve its solution speed, the constraint equation is defined as follows:

$$
\begin{bmatrix} V_{trop} \\ V_{iono} \end{bmatrix} = \begin{bmatrix} 1 & -1 & \underset{1 \times (s+1)}{0} & \underset{1 \times (s+1)}{0} & \underset{1 \times [n \times (s+1)]}{0} \\ 0 & 0 & E & -E & \underset{(s+1) \times [n \times (s+1)]}{0} \end{bmatrix} \begin{bmatrix} RZTD_t \\ RZTD_{t+1} \\ I_{t,t+1} \\ N \end{bmatrix} - \begin{bmatrix} 0 \\ 0 \end{bmatrix}
\tag{8}
$$

where $E$ is the unit matrix.

Since the ionospheric delay error is inversely proportional to its own frequency, and the tropospheric delay error depends on the relative zenith tropospheric delay, the frequencies of the B1I, B1C, B2a, and B3I signals have the same ionospheric and tropospheric parameters. The direct constraint of multi-frequency observation data can be strength-

ened only through constraining the B1I or B1C frequency between epochs. Therefore, the multi-frequency data of the BDS are beneficial to the constraint and solution of ionospheric parameters. The observation equation and constraint conditions are combined, and a parameter elimination method is used in order to eliminate the parameters of the normal equation. The least squares (LSQ) algorithm is employed in order to solve the unknown parameters of Equation (7). Further, the least square ambiguity decorrelation adjustment (LAMBDA) method is applied to the AR. Finally, the double-difference integer ambiguity algebraic sum between the closed reference stations of the same satellite is zero, and *ratio* > 3 is used to test the integer ambiguity results, which improves the reliability of the integer AR results.

### 2.2. Undifferenced Error Correction Value Calculation

After the double-difference ambiguity of the reference station is determined, the double-difference integer ambiguity of frequency *i* in the BDS system is analyzed. The relationship between the first undifferenced integer ambiguity and the corresponding double-difference ambiguity of satellites *p*, *q*, and *k*, where *q* is a reference satellite, at reference stations *A*, *B*, and *C*, is obtained as follows:

$$
\begin{cases}
\Delta\nabla N_{AB,i}^{pq} = N_{A,i}^{p} - N_{A,i}^{q} + N_{B,i}^{q} - N_{B,i}^{p} \\
\Delta\nabla N_{BC,i}^{pq} = N_{B,i}^{p} - N_{B,i}^{q} + N_{C,i}^{q} - N_{C,i}^{p} \\
\Delta\nabla N_{CA,i}^{pq} = N_{C,i}^{p} - N_{C,i}^{q} + N_{A,i}^{q} - N_{A,i}^{p}
\end{cases}
\tag{9}
$$

$$
\begin{cases}
\Delta\nabla N_{AB,i}^{kq} = N_{A,i}^{k} - N_{A,i}^{q} + N_{B,i}^{q} - N_{B,i}^{k} \\
\Delta\nabla N_{BC,i}^{kq} = N_{B,i}^{k} - N_{B,i}^{q} + N_{C,i}^{q} - N_{C,i}^{k} \\
\Delta\nabla N_{CA,i}^{kq} = N_{C,i}^{k} - N_{C,i}^{q} + N_{A,i}^{q} - N_{A,i}^{k}
\end{cases}
\tag{10}
$$

There are only two linearly independent double-difference integer ambiguity values in Equations (9) and (10). Therefore, in order to filter the undifferenced integer ambiguity of a single satellite at a single station, the reference station undifferenced reference integer ambiguity and satellite undifferenced reference integer ambiguity are defined and used to convert double-difference integer ambiguity into undifferenced integer ambiguity. The reference station *A* and satellite *q* are selected for undifferenced reference ambiguity. An integer value can be set as a constant, which will not affect the positioning of a user station. The undifferenced integer ambiguity related to the reference station and reference satellite is set to zero. Then, the ambiguity calculation process of the other satellites is performed as follows:

$$
\begin{cases}
N_{A,i}^{p} = 0 \\
N_{B,i}^{p} = N_{A,i}^{p} - N_{A,i}^{q} + N_{B,i}^{q} - \Delta\nabla N_{AB,i}^{pq} = -\Delta\nabla N_{AB,i}^{pq} \\
N_{C,i}^{p} = N_{B,i}^{p} - N_{B,i}^{q} + N_{C,i}^{q} - \Delta\nabla N_{BC,i}^{pq} = -\Delta\nabla N_{AB,i}^{pq} - \Delta\nabla N_{BC,i}^{pq}
\end{cases}
\tag{11}
$$

$$
\begin{cases}
N_{A,i}^{k} = 0 \\
N_{B,i}^{k} = N_{A,i}^{k} - N_{A,i}^{q} + N_{B,i}^{q} - \Delta\nabla N_{AB,i}^{kq} = -\Delta\nabla N_{AB,i}^{kq} \\
N_{C,i}^{k} = N_{B,i}^{k} - N_{B,i}^{q} + N_{C,i}^{q} - \Delta\nabla N_{BC,i}^{kq} = -\Delta\nabla N_{AB,i}^{kq} - \Delta\nabla N_{BC,i}^{kq}
\end{cases}
\tag{12}
$$

where $N_{A,i}^{p}$, $N_{A,i}^{k}$, $N_{A,i}^{q}$, $N_{B,i}^{q}$, and $N_{C,i}^{q}$ relate to the undifferenced reference ambiguity, and $\Delta\nabla N_{AB,i}^{pq}$, $\Delta\nabla N_{BC,i}^{pq}$, $\Delta\nabla N_{AB,i}^{kq}$, and $\Delta\nabla N_{BC,i}^{kq}$ relate to the known double-difference integer ambiguity.

Through the above process, the double-difference integer ambiguity can be converted into undifferenced integer ambiguity.

After determining the undifferenced integer ambiguity of each satellite corresponding to a particular station, the undifferenced error correction value of each reference station

corresponding to the satellite can be calculated. The inverse distance weighted interpolation algorithm is used to establish a regional high-precision undifferenced error correction model. The undifferenced error correction value of a user station is calculated using the undifferenced error correction value of the reference station and used to correct the error of the user station. The calculation process of the undifferenced error correction value of a satellite $p$ corresponding to the three reference stations $A$, $B$, and $C$ is as follows:

$$
\begin{cases}
Cor^p_{A,i} = \lambda_i \varphi_{A,i} - \rho^p_A + \lambda_i N^p_{A,i} = c(t_A - t^p) + O^p - I^p_{A,i} + T^p - \lambda_i(\sigma_A - \sigma^p) + \varepsilon^p_{A,i} \\
Cor^p_{B,i} = \lambda_i \varphi_{B,i} - \rho^p_B + \lambda_i N^p_{B,i} = c(t_B - t^p) + O^p - I^p_{B,i} + T^p - \lambda_i(\sigma_B - \sigma^p) + \varepsilon^p_{B,i} \\
Cor^p_{C,i} = \lambda_i \varphi_{C,i} - \rho^p_C + \lambda_i N^p_{C,i} = c(t_C - t^p) + O^p - I^p_{C,i} + T^p - \lambda_i(\sigma_C - \sigma^p) + \varepsilon^p_{C,i}
\end{cases} \quad (13)
$$

where $Cor^p_{A,i}$, $Cor^p_{B,i}$, and $Cor^p_{C,i}$ are undifferenced error corrections of satellite $p$ at reference stations $A$, $B$, and $C$, respectively.

According to Equation (13), the undifferenced error correction values mainly address the satellite and receiver clock errors, satellite and receiver hardware delays, ionospheric delay errors, tropospheric delay errors, and orbit errors.

In network RTK, a comprehensive undifferenced error correction value can be directly used for the error correction of a rover station. However, in long-range situations, the correction value obtained via the comprehensive error processing method is not accurate. Therefore, it is necessary to classify the undifferenced error correction values according to the nature of the error. The schematic diagram of the classification region error interpolation method is shown in Figure 2, where $A$, $B$ and $C$ denote reference stations, $U$ is a rover station, and $IPP$ represents an ionospheric puncture point.

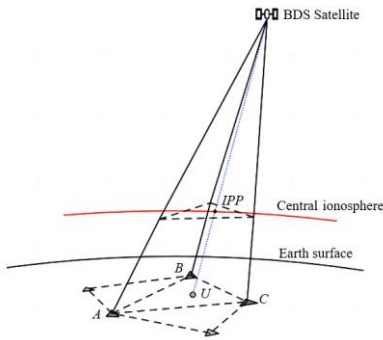

**Figure 2.** The classification error interpolation diagram.

The comprehensive undifferenced error correction value contains mainly undifferenced dispersion error dominated by ionospheric delays and undifferenced non-dispersion error dominated by tropospheric delays. Due to the different error characteristics, it is necessary to classify the comprehensive undifferenced error correction value. The undifferenced dispersion error was calculated using the geometry-free (*GF*) model, and the non-dispersion error was calculated using the ionospheric-free (*IF*) model. The solution process is shown below:

$$
\begin{cases}
GF(Cor^p_{r,ij}) = Cor^p_{r,i} - Cor^p_{r,j} = I^p_{r,j} - I^p_{r,i} + \lambda_i(\sigma_r - \sigma^p) - \lambda_j(\sigma_r - \sigma^p) + \varepsilon^p_{r,i} - \varepsilon^p_{r,j} \\
IF(Cor^p_{r,ij}) = \alpha_{i,i,j}Cor^p_{r,i} - \alpha_{j,i,j}Cor^p_{r,j} = c(t_r - t^p) + T^p + \lambda_i\alpha_{i,i,j}\sigma_r - \lambda_j\alpha_{j,i,j}\sigma_r \\
\qquad\qquad + \lambda_j\alpha_{j,i,j}\sigma^p - \lambda_i\alpha_{i,i,j}\sigma^p + \alpha_{i,i,j}\varepsilon^p_{r,i} - \alpha_{j,i,j}\varepsilon^p_{r,j}
\end{cases} \quad (14)
$$

where $j$ is the other frequency, and $\alpha_{i,i,j} = \frac{f_i^2}{f_i^2 - f_j^2}$, $\alpha_{j,i,j} = \frac{f_j^2}{f_i^2 - f_j^2}$ is the ionosphere-free combination operator. It can be seen from Equation (14) that both dispersion and non-

dispersion error are influenced by other frequency phase hardware delays and observation noise. The error expressions after classification are provided below:

$$\begin{cases} Dis_{r,i}^p = -\alpha_{j,i,j}(Cor_{r,i}^p - Cor_{r,j}^p) \\ UDis_{r,i}^p = \alpha_{i,i,j}Cor_{r,i}^p - \alpha_{j,i,j}Cor_{r,j}^p \\ Cor_{r,i}^p = Dis_{r,i}^p + UDis_{r,i}^p \end{cases} \tag{15}$$

where $Dis_{r,i}^p$ is the undifferenced dispersion error, and $UDis_{r,i}^p$ is the undifferenced non-dispersion error. The undifferenced non-dispersion error is interpolated on the Earth's surface, where the reference and rover stations are located, and the interpolation calculation of the undifferenced dispersion error of the rover station is performed on the interpolation plane at the height of the central ionosphere [24]. It is needed only to calculate the classification error for the reference station, and the classification error correction value of the rover station can be obtained using the inverse distance weighted interpolation algorithm, as follows:

$$Class_{U,i}^p = \begin{bmatrix} a_1 & a_2 & a_3 & b_1 & b_2 & b_3 \end{bmatrix} \begin{bmatrix} Dis_{A,i}^p & Dis_{B,i}^p & Dis_{C,i}^p & UDis_{A,i}^p & UDis_{B,i}^p & UDis_{C,i}^p \end{bmatrix}^T \tag{16}$$

where $Class_{U,i}^p$ is the classification error of a satellite $p$ corresponding to the rover station $U$; $a_1$, $a_2$, and $a_3$ are the weighted coefficients for undifferenced dispersion error of reference stations $A$, $B$, and $C$, respectively; $b_1$, $b_2$, and $b_3$ are the weighted coefficients for the undifferenced non-dispersion error of reference stations $A$, $B$, and $C$, respectively. The weighting coefficient here is obtained through weighting the geometric distance between each base station and the rover station. The equation for calculating the weighting coefficient of the non-dispersive error is as follows:

$$\begin{cases} a_1 = \frac{1/d_{UA}}{1/d_{UA}+1/d_{UB}+1/d_{UC}} \\ a_2 = \frac{1/d_{UB}}{1/d_{UA}+1/d_{UB}+1/d_{UC}} \\ a_3 = \frac{1/d_{UC}}{1/d_{UA}+1/d_{UB}+1/d_{UC}} \end{cases} \tag{17}$$

where $d_{UA}$, $d_{UB}$, and $d_{UC}$ are the geometric distances from stations $A$, $B$, and $C$, respectively, to rover station $U$. The calculation method of the weighting coefficient of the dispersion error is the same as that of the non-dispersion error, but the interpolation plane is in the central ionosphere, and the geometric distance is calculated using the $IPP$ of the station and the satellite.

The classification error correction value of each frequency observation is generated in the same way as the error correction value of frequency $i$. In addition, the interpolation method of the non-dispersive error is consistent with that of the ionospheric delay error, but their interpolation planes are different. The weighting coefficient of the ionospheric delay error is obtained using the location of the puncture point.

### 2.3. Phase Integer AR and Rover Station Positioning

Consider observation $U$ of satellites $p$ and $q$, where satellite $q$ is a reference satellite; then, the phase undifferenced observation equation of a rover station at frequency $i$ is obtained as follows:

$$\begin{cases} L_{U,i}^q = H^q \cdot X + \rho_U^q - \lambda_i N_{U,i}^q + O^q - I_{U,i}^q + T_U^q + ct_U - ct^q - \lambda_i \sigma_U + \lambda_i \sigma^q + \varepsilon_{U,i}^q \\ L_{U,i}^p = H^p \cdot X + \rho_U^p - \lambda_i N_{U,i}^p + O^p - I_{U,i}^p + T_U^p + ct_U - ct^p - \lambda_i \sigma_U + \lambda_i \sigma^p + \varepsilon_{U,i}^p \end{cases} \tag{18}$$

The satellite clock error and satellite hardware delay in an observation can be eliminated after the error correction of a rover station through undifferenced error correction. In

this way, the tropospheric delay error, ionospheric delay error, and satellite orbit error are significantly weakened. The error equation is defined as follows:

$$\begin{cases} L_{U,i}^{q} - OMC_{U,i}^{q} = H^q \cdot X + \rho_U^q - \lambda_i N_{U,i}^q + \Delta ct_U - \Delta\lambda_i\sigma_U \\ L_{U,i}^{p} - OMC_{U,i}^{p} = H^p \cdot X + \rho_U^p - \lambda_i N_{U,i}^p + \Delta ct_U - \Delta\lambda_i\sigma_U \end{cases} \tag{19}$$

where $OMC_{U,i}^{q}$ and $OMC_{U,i}^{p}$ are the sums of the undifferenced error correction values of the rover station obtained at multiple reference stations; $\Delta ct_U$ is the residual receiver clock error; and $\Delta\lambda_i\sigma_U$ denotes the residual of the carrier phase hardware delay at the receiver side.

In order to eliminate the residual error of the receiver clock error and receiver hardware delay, the inter-satellite difference of Equation (19) can be obtained via:

$$\begin{aligned} V_{U,i}^{pq} &= (-H^p + H^q) \cdot X + \lambda_i(N_{U,i}^p - N_{U,i}^q) \\ &\quad + \left\{ [(L_{U,i}^p - OMC_{U,i}^p) - (L_{U,i}^q - OMC_{U,i}^q)] - (\rho_U^p - \rho_U^q) \right\} \end{aligned} \tag{20}$$

where,

$$\begin{cases} h_{U,i}^{pq} = -H^p + H^q \\ L_{U,i}^{pq} = \left\{ [(L_{U,i}^p - OMC_{U,i}^p) - (L_{U,i}^q - OMC_{U,i}^q)] - (\rho_U^p - \rho_U^q) \right\} \end{cases} \tag{21}$$

The error equation of the phase ambiguity and position parameters of the rover station is defined as:

$$V_{U,i}^{pq} = h_{U,i}^{pq} \cdot X + \lambda_i(N_{U,i}^p - N_{U,i}^q) + L_{U,i}^{pq} \tag{22}$$

Suppose that the BDS-3 multi-frequency observation data relate to $(s + 1)$ satellites' data (the reference satellite number is "1") collected at a rover station; then, the matrix form of the error equation is defined as follows:

$$\begin{bmatrix} V_{U,1}^S \\ \vdots \\ V_{U,n}^S \end{bmatrix} = \begin{bmatrix} H^S & \lambda^S \end{bmatrix} \begin{bmatrix} X \\ N^S \end{bmatrix} + \begin{bmatrix} L^S \end{bmatrix} \tag{23}$$

where,

$$V_{U,i}^S = \begin{bmatrix} V_{U,i}^{S_{1,2}} \\ \vdots \\ V_{U,i}^{S_{1,s+1}} \end{bmatrix}_{s \times 1} \quad H^S = \begin{bmatrix} l_1^{S_{1,2}} & m_1^{S_{1,2}} & e_1^{S_{1,2}} \\ \vdots & \vdots & \vdots \\ l_1^{S_{1,s+1}} & m_1^{S_{1,s+1}} & e_1^{S_{1,s+1}} \\ \vdots & \vdots & \vdots \\ l_n^{S_{1,s+1}} & m_n^{S_{1,s+1}} & e_n^{S_{1,s+1}} \end{bmatrix}_{(n \times s) \times 3} \quad \lambda^S = \begin{bmatrix} \lambda_1^S b & \cdots & 0 \\ \vdots & \ddots & \vdots \\ 0 & \cdots & \lambda_n^S b \end{bmatrix}_{(n \times s) \times (n \times (s+1))} \quad X = \begin{bmatrix} \delta x \\ \delta y \\ \delta z \end{bmatrix}_{3 \times 1} \quad N^S = \begin{bmatrix} N_1^{S_{1,1}} \\ \vdots \\ N_1^{S_{1,s+1}} \\ \vdots \\ N_n^{S_{1,s+1}} \end{bmatrix}_{[n \times (s+1)] \times 1} \quad L^S = \begin{bmatrix} L_{U,1}^{S_{1,2}} \\ \vdots \\ L_{U,1}^{S_{1,s+1}} \\ \vdots \\ L_{U,n}^{S_{1,s+1}} \end{bmatrix}_{(n \times s) \times 1}$$

where $l$, $m$, and $e$ are the coefficients of position parameters; superscript $S$ represents different systems; subscript $n$ denotes the maximum number of frequencies.

After the undifferenced error correction at the rover station, compared to the reference station, the unknown parameters of the rover station include three additional position parameters and integer ambiguity parameters. The float solution of the ambiguity of the rover station is obtained using the LSQ method of parameter elimination. Then, it is introduced in the LAMBDA for AR. Finally, the integer ambiguity is substituted into the observation equation in order to obtain the position of the rover station.

## 3. Experimental Results Analysis

In this study, two sets of long baseline observation data were solved using the self-made long-range URTK positioning algorithm program. The experiment included BDS-3 single-system B1C/B2a and B1I/B3I dual-frequency signals, B1C/B2a/B3I and

B1I/B2a/B3I triple-frequency signals, B1I/B1C/B2a/B3I four-frequency signals, and GPS single-system L1/L2 dual-frequency six-combination mode signals. The experiment was conducted using the CORS network observation data of a province collected on 20 August 2022. The sampling interval was 1 s, the observation time was 24 h, and the satellite elevation angle was set to 15°. The installation diagram of a data acquisition device in the CORS network is shown in Figure 3, where it can be seen that it is mainly composed of the GNSS receiver and GNSS antenna.

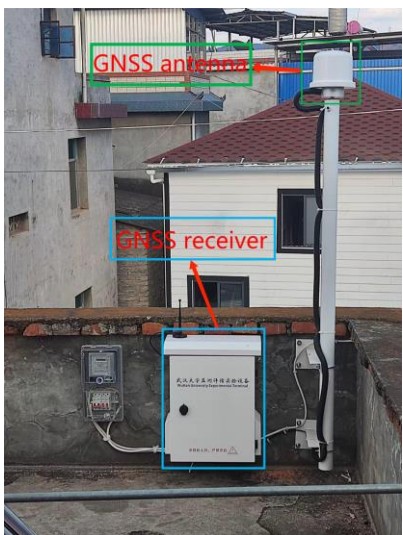

**Figure 3.** Data acquisition equipment.

　　The observation data related to four reference stations denoted as *A*, *B*, *C*, and *D* and two rover stations denoted as $U_1$ and $U_2$. The distances between stations are given in Table 1, where it can be seen that the lengths of baselines *A-B*, *B-C*, *C-A*, *A-D*, and *D-C* were 108 km, 136 km, 151 km, 161 km, and 172 km, respectively, and the distance between the two rover stations was 85 km. In this study, the reference stations were divided into two groups: reference stations of network 1 (*A*, *B*, *C*) and reference stations of network 2 (*A*, *D*, *C*). The station azimuth distribution is presented in Figure 4. All experiments were conducted using the same data.

**Table 1.** Baseline information.

| Baseline | *A-B* | *B-C* | *C-A* | *A-D* | *D-C* | $U_1$-$U_2$ |
|---|---|---|---|---|---|---|
| Length (km) | 108 | 136 | 151 | 161 | 172 | 85 |

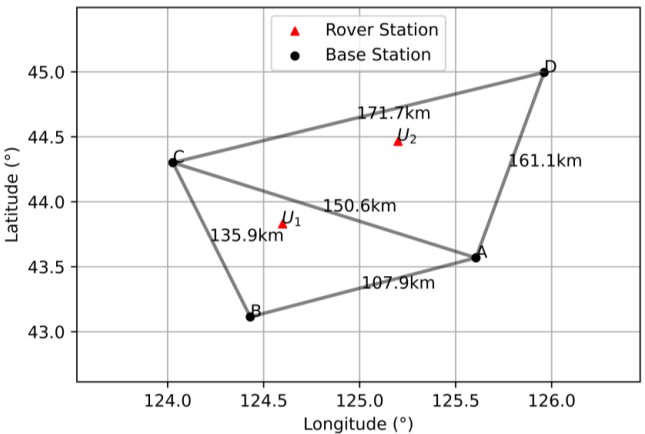

**Figure 4.** Distribution of stations.

The two rover stations were located in the same region, and the satellite visibility at the two stations was similar. The satellite visibility of rover station $U_1$ is presented in Figure 5. The number of B1I/B1C/B2a/B3I, B1C/B2a/B3I, B1I/B2a/B3I, and B1C/B2a observation data satellites of the BDS-3 was approximately eight; the number of B1I/B3I observation data satellites, nine, was the largest; and the number of GPS satellites was seven, which could meet the calculation requirements.

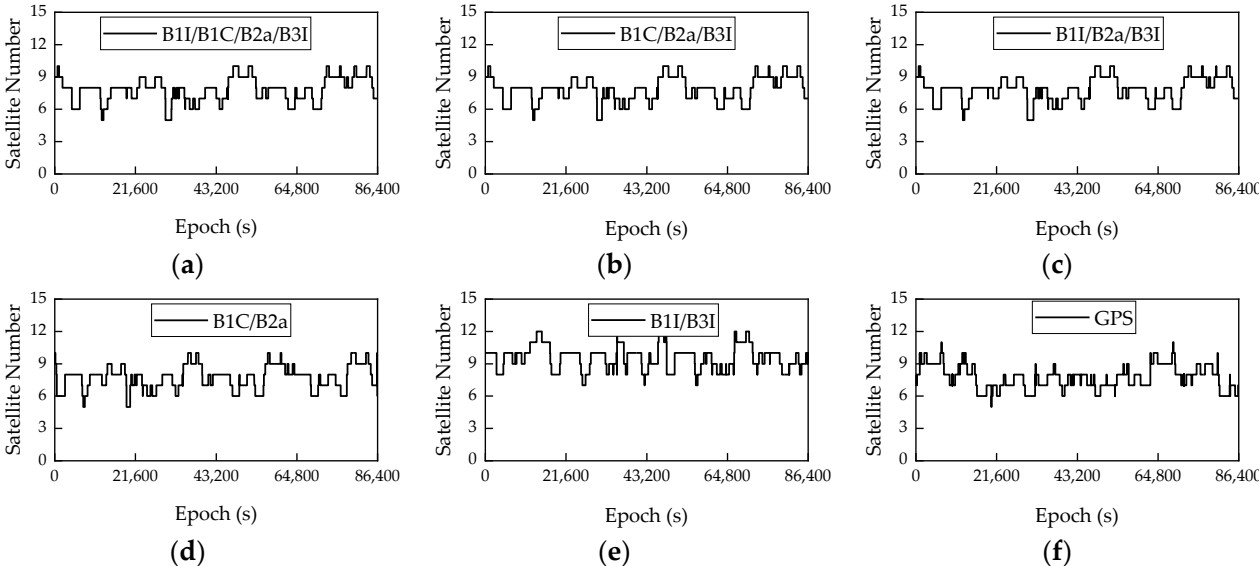

**Figure 5.** (**a**) The number of satellites at rover station $U_1$ for B1I/B1C/B2a/B3I. (**b**) The number of satellites at rover station $U_1$ for B1C/B2a/B3I. (**c**) The number of satellites at rover station $U_1$ for B1I/B2a/B3I. (**d**) The number of satellites at rover station $U_1$ for B1C/B2a. (**e**) The number of satellites at rover station $U_1$ for B1I/B3I. (**f**) The number of satellites at rover station $U_1$ for GPS.

The position dilution of precision (PDOP) of each frequency combination was calculated using the observation data and broadcast ephemeris of $U_1$ and $U_2$, as shown in Figure 6. The PDOP indicated the position accuracy, which specifically represents the spatial geometric strength of satellite distribution. The smaller the PDOP value was, the better the satellite distribution was. Generally, a PDOP value of less than three was an ideal state [33]. In Figure 6, it can be clearly seen that the PDOP values of B1I/B1C/B2a/B3I, B1C/B2a/B3I, and B1I/B2a/B3I were basically the same. Although the PDOP values of B1C/B2a, B1I/B3I, and GPS showed different trends, all PDOP values were lower than two. Thus, all combinations met the ideal state required for positioning. In order to compare the PDOP values of several combinations in more detail, the PDOP values were statistically analyzed, and the maximum, minimum, and average PDOP values of B1I/B1C/B2a/B3I, B1C/B2a, B1I/B3I, and GPS were calculated, as shown in Table 2. Also shown in Table 2, the maximum and minimum PDOP values of B1I/B1C/B2a/B3I and B1C/B2a were almost the same, and their average PDOP values differed slightly, which was consistent with the results presented in Figure 6. Although the trends of B1I/B3I and GPS were completely different from the former, the numerical difference was very small. The PDOP values of different combinations ranged from 0.77 to 1.96, with an average value of approximately 1.1. Therefore, different frequency combinations of the BDS-3 signals could meet the same requirements as the GPS signals in terms of satellite number and PDOP value.

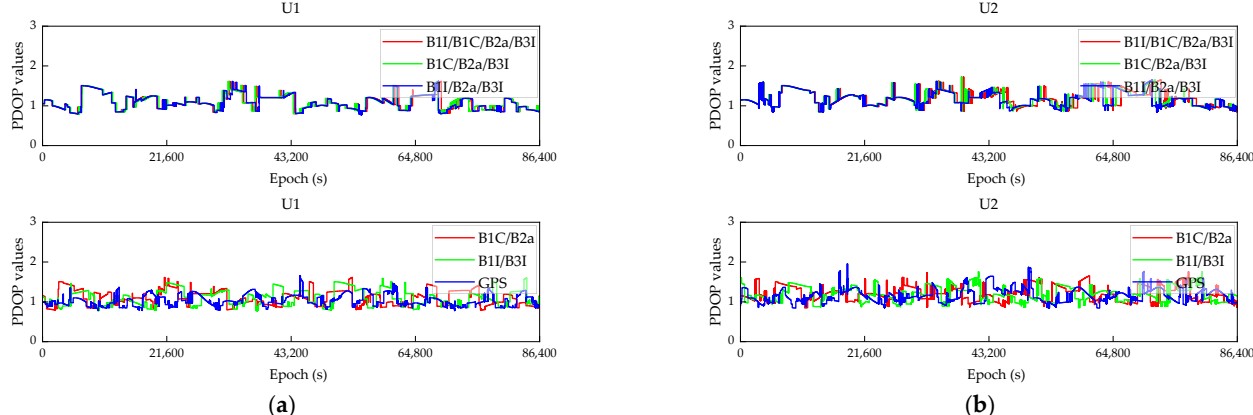

**Figure 6.** (**a**) The PDOP values of rover station $U_1$ and (**b**) the PDOP values of rover station $U_2$.

**Table 2.** The PDOP statistical results.

| Combined Model | Rover Station | Maximum Value | Minimum Value | Mean Value |
|---|---|---|---|---|
| B1I/B1C/B2a/B3I | $U_1$ | 1.616 | 0.794 | 1.093 |
| | $U_2$ | 1.725 | 0.847 | 1.191 |
| B1C/B2a | $U_1$ | 1.616 | 0.787 | 1.102 |
| | $U_2$ | 1.761 | 0.847 | 1.209 |
| B1I/B3I | $U_1$ | 1.605 | 0.778 | 1.102 |
| | $U_2$ | 1.760 | 0.848 | 1.196 |
| GPS | $U_1$ | 1.658 | 0.783 | 1.030 |
| | $U_2$ | 1.958 | 0.841 | 1.158 |

### 3.1. Reference Station AR

The accurate resolution of the integer ambiguity of a reference station has been the main challenge in network RTK positioning, and it is crucial to realize the comprehensive undifferenced error correction via converting double-difference ambiguity into undifferenced ambiguity. Therefore, before the ambiguity was fixed, it was necessary to obtain an accurate ambiguity float solution in order to achieve rapid and accurate AR. Figures 7 and 8 show the convergence process of B1C/B2a and GPS double-difference ambiguity float solutions into the baseline solution of reference station network 1 (*A*, *B*, *C*), respectively.

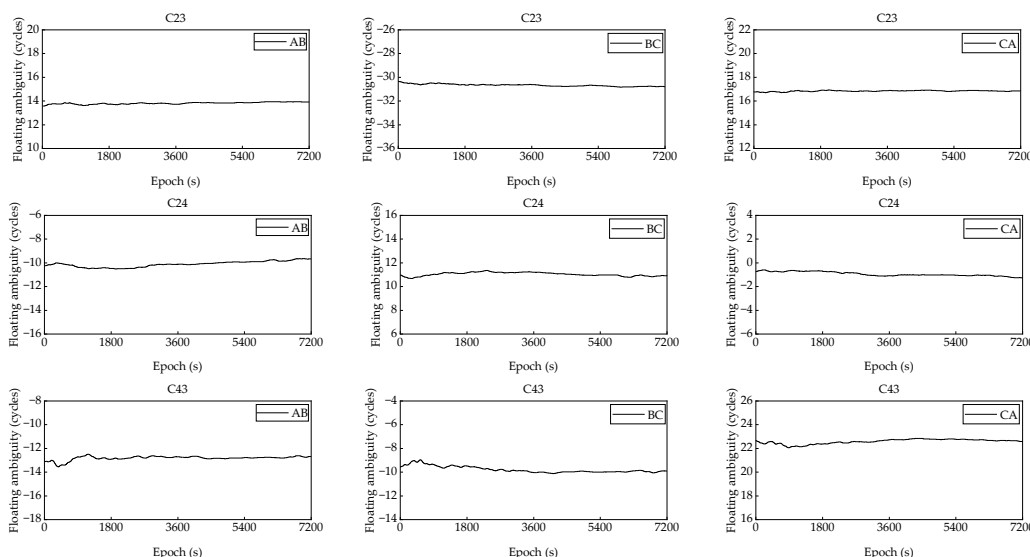

**Figure 7.** The convergence process of B1C/B2a partial satellite float ambiguity.

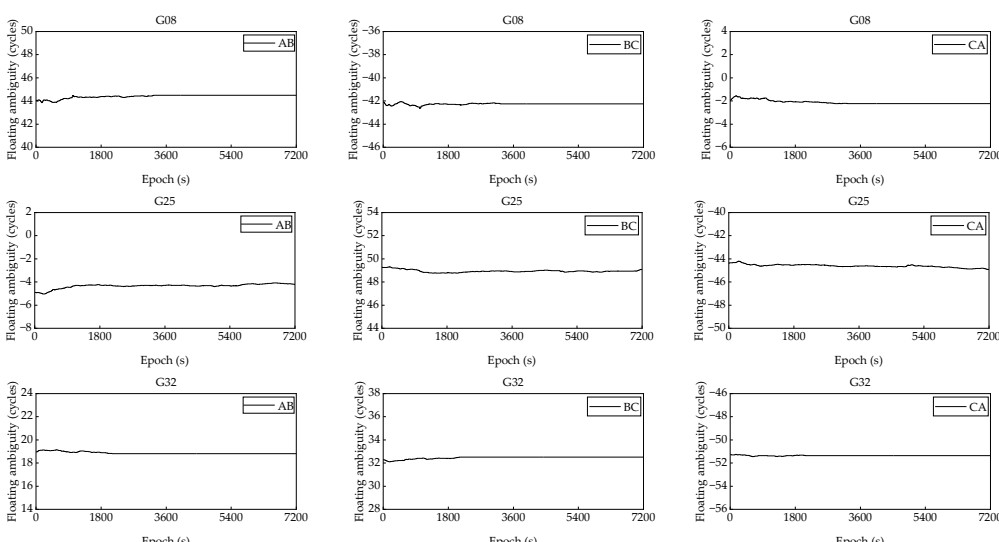

**Figure 8.** The convergence process of GPS partial satellite float ambiguity.

Due to the estimation of the residual atmospheric delay error, for the convergence process of the double-difference ambiguity float solution of each satellite, the ambiguity float solution was rapidly stabilized near the accurate integer ambiguity. In addition, the double-difference ambiguity of the three baselines of the same satellite basically satisfied the closed condition that the sum was zero. When the LAMBDA was used to search for fixed solutions, the calculated value was small, and the fixation success rate was higher. The ratio values of rover stations $U_1$ and $U_2$ are shown in Figure 9, where it can be seen that the ratio values of B1I/B1C/B2a/B3I, B1C/B2a/B3I, and B1I/B2a/B3I were high, and their trends were basically the same. The ratio value of B1I/B3I in dual-frequency mode was the smallest among all combinations, and the GPS performed better than B1C/B2a and B1I/B3I in most time periods. The ratio values in some of the epochs increased because of the emergence of new satellites or the disappearance of individual satellites. When the reference satellite changed, for the satellite whose ambiguity was fixed, the reference satellite transformation could be performed on the current arc observation value, continuing to use the fixed ambiguity, and the double-difference phase integer AR in subsequent epochs was obtained. The sum of double-difference ambiguity values between closed reference stations was zero, as one of the ambiguity-related constraints, which was added to the observation equation of the AR in order to improve the efficiency and success rate of the ambiguity search.

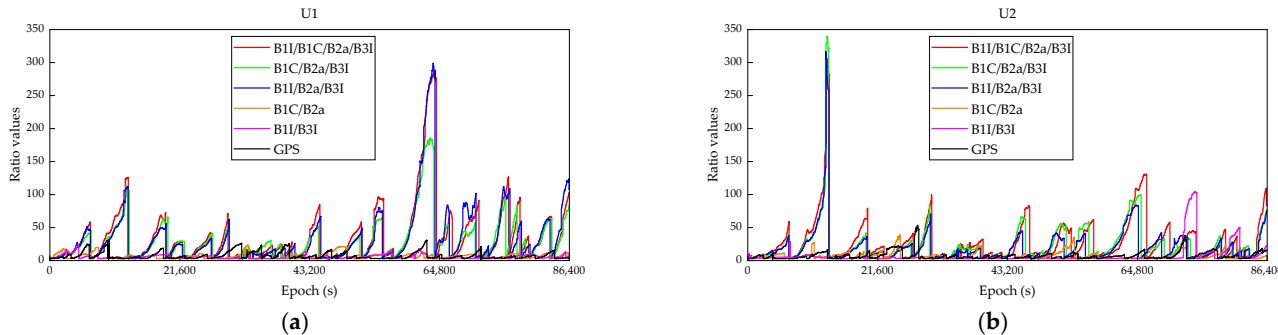

**Figure 9.** (**a**) Ratio values of rover station $U_1$ and (**b**) ratio values of rover station $U_2$.

Tables 3–6 show the success rate results for the AR of some of the satellites in the two reference station networks of B1I/B1C/B2a/B3I, B1C/B2a/B3I, B1C/B2a, and GPS. As shown in Tables 3–6, the AR success rates of the two reference station networks were high, and reference station network 1 performed better than reference station network 2.

The B1I/B1C/B2a/B3I and B1C/B2a/B3I had better AR rates than did the dual-frequency data, and they were almost all above 99%. The B1C/B2a had the same AR rates as the GPS, most of which were nearly 98%. Thus, compared to the dual-frequency data, the multi-frequency data had more advantages in their AR success rate. This could be due to the fact that multi-frequency data could provide more accurate ambiguity alternatives when the LAMBDA ambiguity search was fixed.

**Table 3.** Statistical results of the B1I/B1C/B2a/B3I AR success rate.

| PRN | Reference Station Network | Number of Observations | Fixed Number | Unfixed Number | Fixed Success Rate (%) |
|-----|---------------------------|------------------------|--------------|----------------|------------------------|
| C22 | 1 | 17,464 | 17,389 | 75 | 99.57 |
|     | 2 | 14,878 | 14,739 | 139 | 99.06 |
| C27 | 1 | 18,035 | 17,851 | 184 | 98.97 |
|     | 2 | 18,035 | 18,035 | 0 | 100 |
| C32 | 1 | 22,195 | 21,998 | 197 | 99.11 |
|     | 2 | 22,195 | 22,195 | 0 | 100 |
| C35 | 1 | 8135 | 8135 | 0 | 100 |
|     | 2 | 8135 | 8135 | 0 | 100 |
| C45 | 1 | 20,823 | 20,823 | 0 | 100 |
|     | 2 | 20,823 | 20,586 | 237 | 98.86 |

**Table 4.** Statistical results of the B1C/B2a/B3I AR success rate.

| PRN | Reference Station Network | Number of Observations | Fixed Number | Unfixed Number | Fixed Success Rate (%) |
|-----|---------------------------|------------------------|--------------|----------------|------------------------|
| C22 | 1 | 17,464 | 17,385 | 79 | 99.54 |
|     | 2 | 14,878 | 14,863 | 15 | 99.89 |
| C27 | 1 | 18,035 | 17,856 | 179 | 99.01 |
|     | 2 | 18,035 | 18,035 | 0 | 100 |
| C32 | 1 | 22,195 | 22,120 | 75 | 99.66 |
|     | 2 | 22,195 | 22,195 | 0 | 100 |
| C35 | 1 | 8135 | 8135 | 0 | 100 |
|     | 2 | 8135 | 8135 | 0 | 100 |
| C45 | 1 | 20,823 | 20,738 | 85 | 99.59 |
|     | 2 | 20,823 | 20,573 | 250 | 98.79 |

**Table 5.** Statistical results of the B1C/B2a AR success rate.

| PRN | Reference Station Network | Number of Observations | Fixed Number | Unfixed Number | Fixed Success Rate (%) |
|-----|---------------------------|------------------------|--------------|----------------|------------------------|
| C22 | 1 | 17,464 | 17,123 | 341 | 98.04 |
|     | 2 | 14,878 | 14,464 | 414 | 97.21 |
| C27 | 1 | 18,035 | 17,752 | 283 | 98.43 |
|     | 2 | 18,035 | 17,519 | 516 | 97.13 |
| C32 | 1 | 22,195 | 21,876 | 319 | 98.56 |
|     | 2 | 22,195 | 21,952 | 243 | 98.90 |
| C35 | 1 | 8135 | 8135 | 0 | 100 |
|     | 2 | 8135 | 8135 | 0 | 100 |
| C45 | 1 | 20,823 | 20,443 | 380 | 98.17 |
|     | 2 | 20,823 | 20,298 | 525 | 97.47 |

**Table 6.** Statistical results of the GPS AR success rate.

| PRN | Reference Station Network | Number of Observations | Fixed Number | Unfixed Number | Fixed Success Rate (%) |
|---|---|---|---|---|---|
| G06 | 1 | 18,413 | 18,234 | 179 | 99.02 |
| | 2 | 16,675 | 16,299 | 376 | 97.74 |
| G13 | 1 | 15,248 | 15,186 | 62 | 99.59 |
| | 2 | 14,808 | 14,702 | 106 | 99.28 |
| G20 | 1 | 20,926 | 20,877 | 49 | 99.76 |
| | 2 | 19,084 | 18,551 | 533 | 97.20 |
| G21 | 1 | 14,860 | 14,060 | 254 | 98.29 |
| | 2 | 14,860 | 14,534 | 326 | 97.80 |
| G29 | 1 | 24,459 | 24,155 | 304 | 98.75 |
| | 2 | 24,459 | 23,885 | 574 | 97.65 |

### 3.2. Undifferenced Error Correction Results

Next, the double-difference integer ambiguities were converted into undifferenced integer ambiguities, and then the undifferenced integer ambiguities were substituted into the observation equation of the reference station in order to obtain the comprehensive undifferenced error correction values corresponding to each reference station. The comprehensive undifferenced error corrections of C25, C39, and G24 satellites in the first frequency continuous visible period of reference stations *A*, *B*, *C*, and *D* were calculated, as shown in Figure 10. The results indicated that the comprehensive undifferenced error correction values of each satellite changed linearly, but the changing trends of the same satellite on the four reference stations were similar.

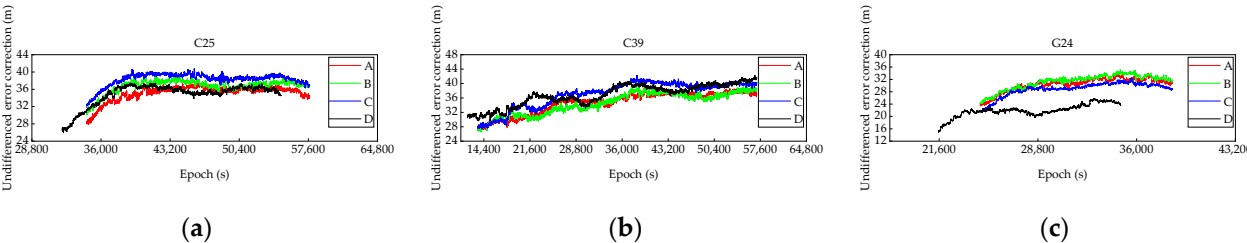

(**a**)  (**b**)  (**c**)

**Figure 10.** (**a**) Comprehensive undifferenced error correction values of the C25 satellite; (**b**) comprehensive undifferenced error correction values of the C39 satellite; and (**c**) comprehensive undifferenced error correction values of the G24 satellite.

For a long-range network RTK, the weakening of atmospheric delay error correlation could reduce the accuracy of positioning results. Therefore, in order to ensure the reliability of the undifferenced corrections obtained via the inverse distance weighted interpolation algorithm, the comprehensive undifferenced error corrections were classified. The comprehensive undifferenced error corrections of two frequencies (B1/B2, B1/B3, or L1/L2) of each satellite were selected. The undifferenced ionospheric delay was calculated using the GF model, and the non-dispersion error was calculated using the IF model. Figures 11–13 show the error calculation results of C21, C25 (MEO satellite), C39, and C40 (IGSO satellite) in the BDS-3, and G10 and G24 (MEO satellite) in the GPS at four reference stations. Because different types of satellites were in different orbits, the satellite's visible time also differed among the orbits. As shown in Figures 11–13, the undifferenced ionospheric delay error of each satellite changed slightly between epochs; the variation trends of the undifferenced ionospheric delay error of the four reference stations for the same satellite were basically the same, and they were proportional to the reciprocal of the height–angle–sine function. The non-dispersion error contained most of the errors in the comprehensive undifferenced error correction process, so its changing trend coincided with that of the comprehensive undifferenced error correction performance. The starting time of station *D* was different

from those of the other three stations, which was because the distance between stations *A* and *D* was 242 km.

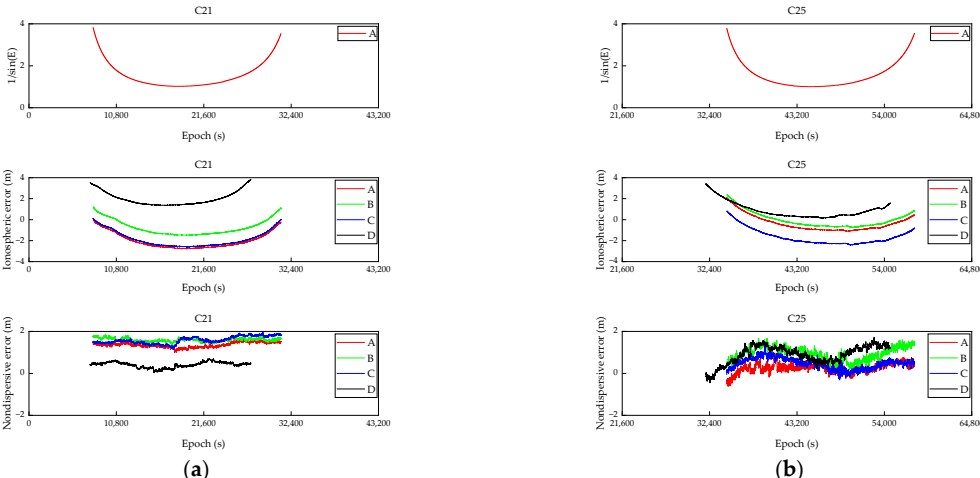

**Figure 11.** (**a**) Classification undifferenced error correction results of the C21 satellite and (**b**) classification undifferenced error correction results of the C25 satellite.

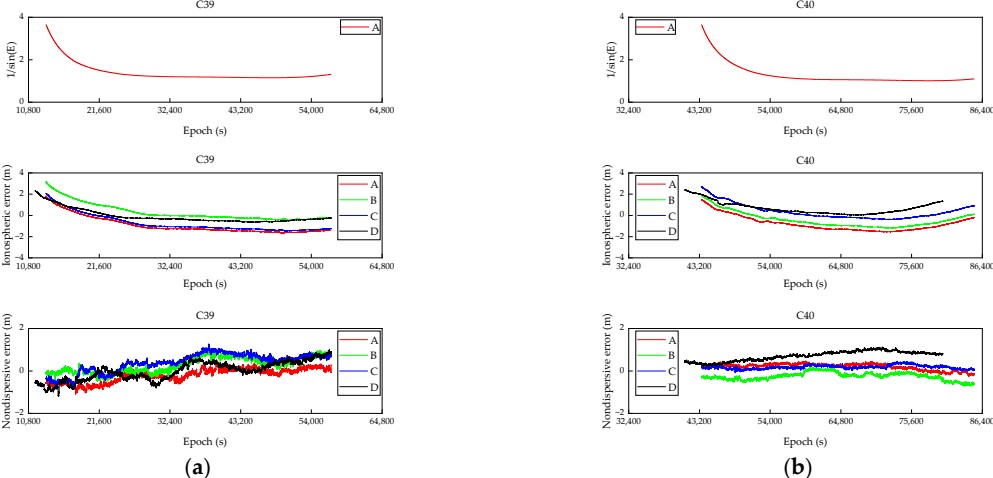

**Figure 12.** (**a**) Classification undifferenced error correction results of the C39 satellite and (**b**) classification undifferenced error correction results of the C40 satellite.

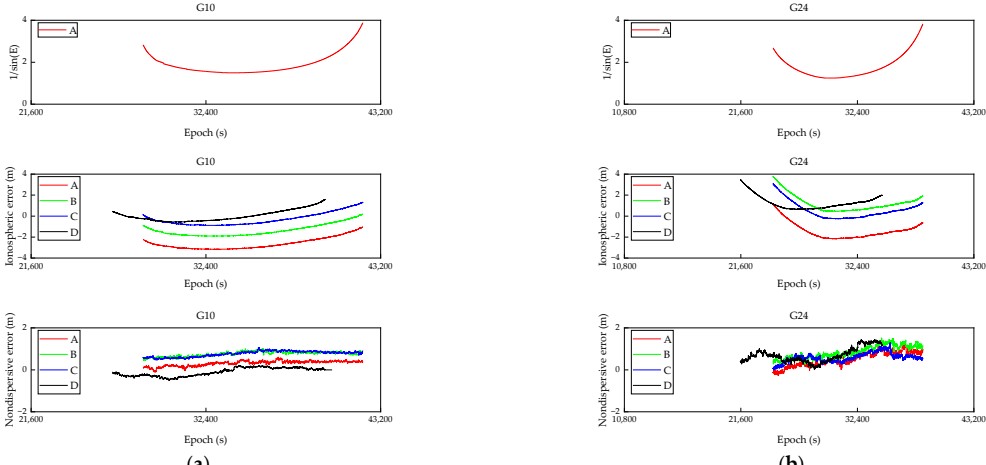

**Figure 13.** (**a**) Classification undifferenced error correction results of the G10 satellite and (**b**) classification undifferenced error correction results of the G24 satellite.

### 3.3. Positioning Accuracy

The undifferenced error correction values were substituted into the observation equation of the rover station, and the accurate position of the current station was obtained after fixing the integer ambiguity. Before analyzing the positioning results, the first convergence time in each combined positioning process is presented in Table 7, where it can be seen that the convergence times of each combination of rover stations $U_1$ and $U_2$ differed slightly. The convergence times of B1I/B1C/B2a/B3I, B1C/B2a/B3I, B1I/B2a/B3I, and B1C/B2a were 2–3 min, while the convergence times of B1I/B3I and GPS were obviously longer, about 6 min, which was twice the former value. Thus, the convergence time of multi-frequency data was shorter, the convergence time of B1C/B2a was better than that of B1I/B3I, and the convergence time of B1I/B3I was slightly worse than that of the GPS.

**Table 7.** Statistical results of the first convergence time(s).

| Combined Model | Rover Station $U_1$ | Rover Station $U_2$ |
|---|---|---|
| B1I/B1C/B2a/B3I | 124 | 164 |
| B1C/B2a/B3I | 122 | 113 |
| B1I/B2a/B3I | 128 | 171 |
| B1C/B2a | 178 | 117 |
| B1I/B3I | 359 | 400 |
| GPS | 286 | 338 |

The positioning result deviations of observation data corresponding to various combination modes of rover station $U_1$ are provided in Figure 14. Considering the number of satellites, although B1I/B3I had the largest number of satellites, its positioning result was not the best. This is mainly due to the improved modulation of the new B1C/B2a frequency, which has better signal quality compared to B1I/B3I [26,27]. However, the positioning results of B1I/B3I here are significantly better than other combinations in the U direction, which is related to the observation environment and the quality of the observation data. This is a special case and will not be discussed in this paper. The positioning result of multi-frequency data was obviously better than that of dual-frequency data, particularly in the elevation direction. The increase in frequency improved not only the success rate of the AR and convergence time, but also the stability of positioning results, and the deviations in the results of multi-frequency data were slighter.

The positioning result deviations in the observation data of various combination modes of rover station $U_2$ are shown in Figure 15. With the increase in the distance between reference stations, the accuracy of network RTK data decreased, but the overall trend was the same as that of rover station $U_1$; the only difference was that the accuracy of B1I/B3I decreased significantly. With the increase in the distance between stations, the correlation of errors, such as atmospheric delay, was weakened. The satellite observation data of low elevation angle had a great influence on the AR, which affected the final positioning result.

The RMS statistical results of the positioning deviations of each combination mode are presented in Table 8. Based on the time series and statistical results of the positioning error, the systematic errors of the BDS-3 and GPS observations were basically eliminated. In the dual-frequency combination, the positioning result of B1C/B2a was better than that of B1I/B3I in the horizontal direction, and the results in the elevation direction were different due to the influence of the distance between the reference stations; namely, B1C/B2a performed better than GPS, and B1I/B3I achieved a result equivalent to that of the GPS. Compared to the GPS, B1C/B2a showed average increases of 10.96%, 40.77%, and 13.73% in the E, N, and U directions, respectively, and performed the same as B1I/B2a/B3I. For the multi-frequency combination, the positioning results of B1C/B2a/B3I and B1I/B2a/B3I were better than those of the dual-frequency combination. Compared to BIC/B2a and B1I/B3I, B1C/B2a/B3I and B1I/B2a/B3I had average increases of 24.69%, 20.41%, and 12.58% in the E, N, and U directions, respectively. Further, compared to the GPS, B1C/B2a/B3I and B1I/B2a/B3I had average increases of 16.16%, 42.73%, and 28.44% in E, N, and U directions,

respectively. Additionally, B1C/B2a/B3I was superior to B1I/B2a/B3I, with average increases of 14.14%, 20.53%, and 10.89% in the E, N, and U directions, respectively. Thus, the B1I/B1C/B2a/B3I positioning results were the best among all combinations. Compared with the dual-frequency combination, the average increases for B1I/B1C/B2a/B3I in the E, N, and U directions were 27.37%, 31.91%, and 26.56%, respectively. In addition, compared with the triple-frequency combination, the average increases for B1I/B1C/B2a/B3I in the E, N, and U directions were 9.38%, 20.75%, and 18.72%, respectively. The positioning results of B1I/B1C/B2a/B3I were significantly better than those of the GPS, with average increases of 26.43%, 70.71%, and 51.86% in the E, N, and U directions, respectively. Compared with the dual-frequency data, the multi-frequency data achieved a superior improvement in positioning accuracy.

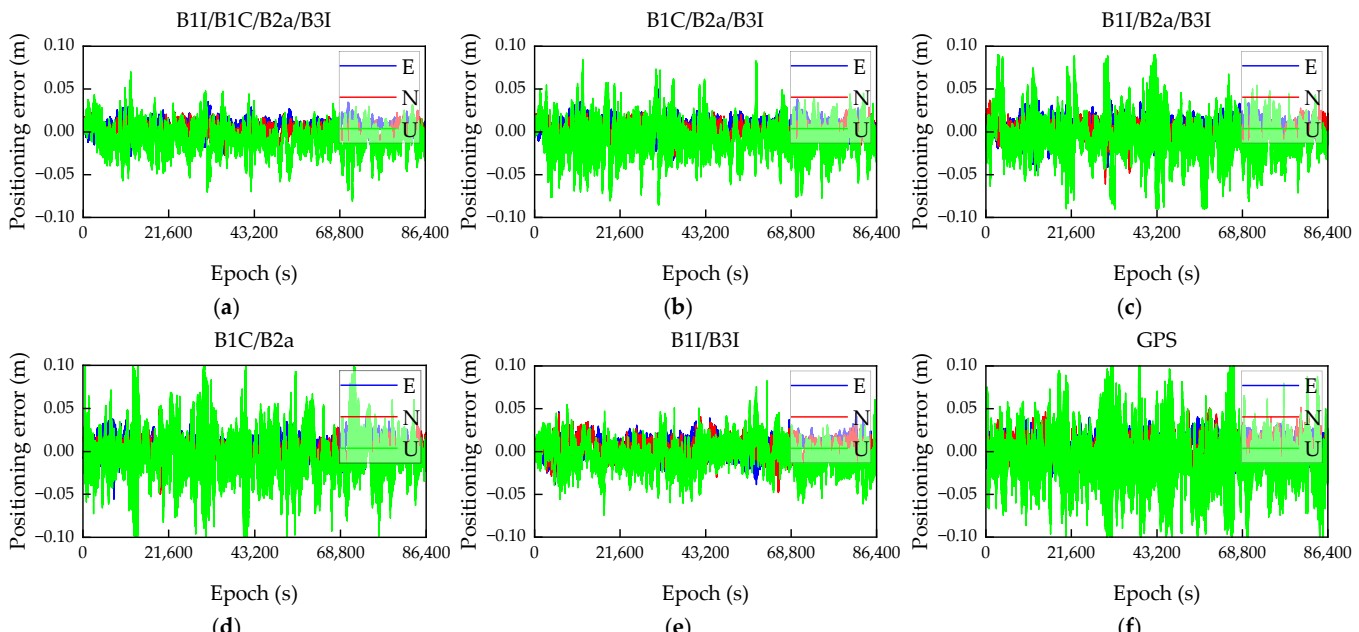

**Figure 14.** (**a**) B1I/B1C/B2a/B3I combined mode positioning deviation results of rover station $U_1$; (**b**) B1C/B2a/B3I combined mode positioning deviation results of rover station $U_1$; (**c**) B1I/B2a/B3I combined mode positioning deviation results of rover station $U_1$; (**d**) B1C/B2a combined mode positioning deviation results of rover station $U_1$; (**e**) B1I/B3I combined mode positioning deviation results of rover station $U_1$; and (**f**) GPS positioning deviation results of rover station $U_1$.

**Table 8.** The RMS values of the positioning deviations of the two rover stations (m).

| Combined Model | Rover Station | E | N | U |
|---|---|---|---|---|
| B1I/B1C/B2a/B3I | $U_1$ | 0.0085 | 0.0075 | 0.0181 |
|  | $U_2$ | 0.0108 | 0.0079 | 0.0227 |
| B1C/B2a/B3I | $U_1$ | 0.0087 | 0.0081 | 0.0206 |
|  | $U_2$ | 0.0110 | 0.0089 | 0.0252 |
| B1I/B2a/B3I | $U_1$ | 0.0096 | 0.0094 | 0.0243 |
|  | $U_2$ | 0.0130 | 0.0103 | 0.0261 |
| B1C/B2a | $U_1$ | 0.0098 | 0.0087 | 0.0249 |
|  | $U_2$ | 0.0122 | 0.0101 | 0.0292 |
| B1I/B3I | $U_1$ | 0.0103 | 0.0099 | 0.0169 |
|  | $U_2$ | 0.0174 | 0.0122 | 0.0330 |
| GPS | $U_1$ | 0.0108 | 0.0126 | 0.0285 |
|  | $U_2$ | 0.0136 | 0.0138 | 0.0332 |

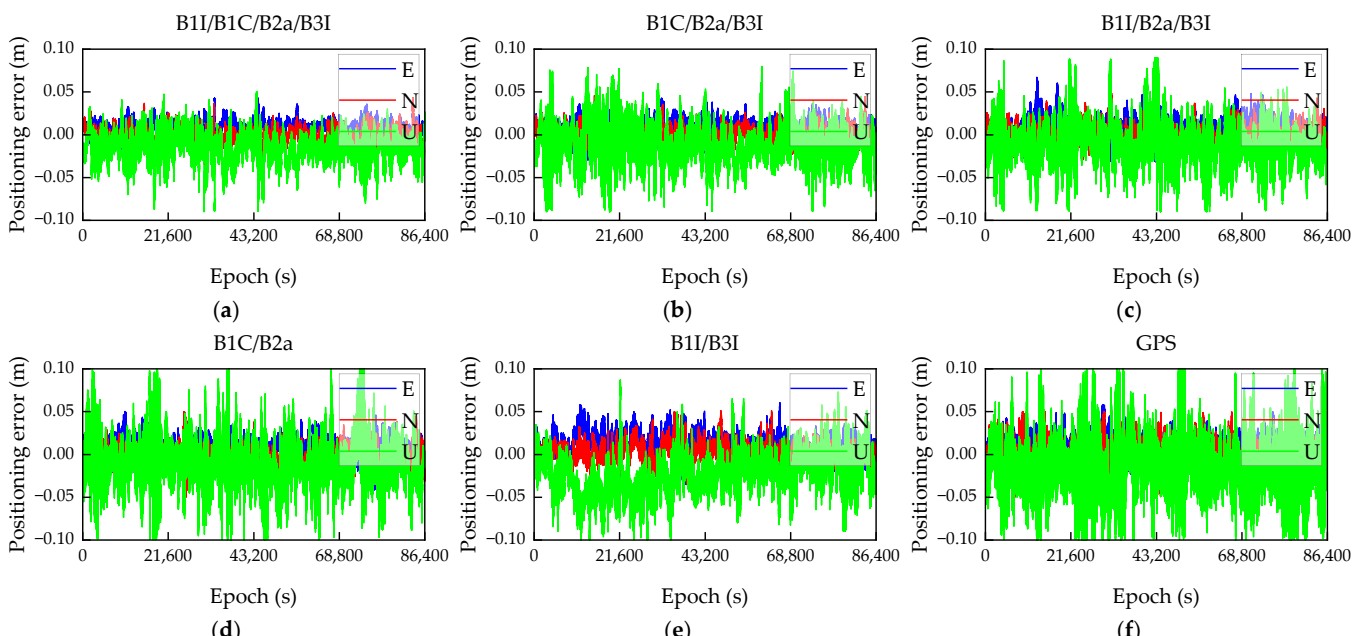

**Figure 15.** (**a**) B1I/B1C/B2a/B3I combined mode positioning deviation results of rover station $U_2$; (**b**) B1C/B2a/B3I combined mode positioning deviation results of rover station $U_2$; (**c**) B1I/B2a/B3I combined mode positioning deviation results of rover station $U_2$; (**d**) B1C/B2a combined mode positioning deviation results of rover station $U_2$; (**e**) B1I/B3I combined mode positioning deviation results of rover station $U_2$; and (**f**) GPS positioning deviation results of rover station $U_2$.

In order to test the positioning results, the residual values of the satellite positioning results of two sets from rover station $U_1$ and rover station $U_2$, corresponding to different frequencies, are shown in Figures 16 and 17. Theoretically, the residual value is close to zero, and in this experiment, it was obtained via substituting the final positioning result and estimated error parameters obtained in the intermediate process into the rover observation equation. The blank segments in Figures 16 and 17 are due to a failure to observe the satellite during this period. As shown in Figures 15 and 16, the residual values of the BDS-3 and GPS were generally near zero. Although the residual values were different because of the different observation times of the satellites, they could be maintained within 1 cm, which conformed to the variation law of residual values and coincided well with the final positioning results. From the perspective of frequency, the B2a residual value of the BDS-3 was the smallest, and it was followed by those of B3I, B1I, and B1C, which were generally equivalent; the L2-frequency residual value of the GPS was significantly better than that of the L1 frequency.

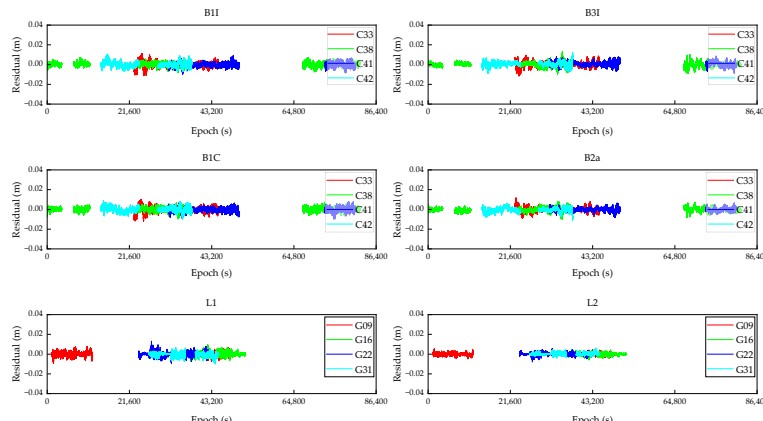

**Figure 16.** The residual values corresponding to different frequencies of satellites for rover station $U_1$.

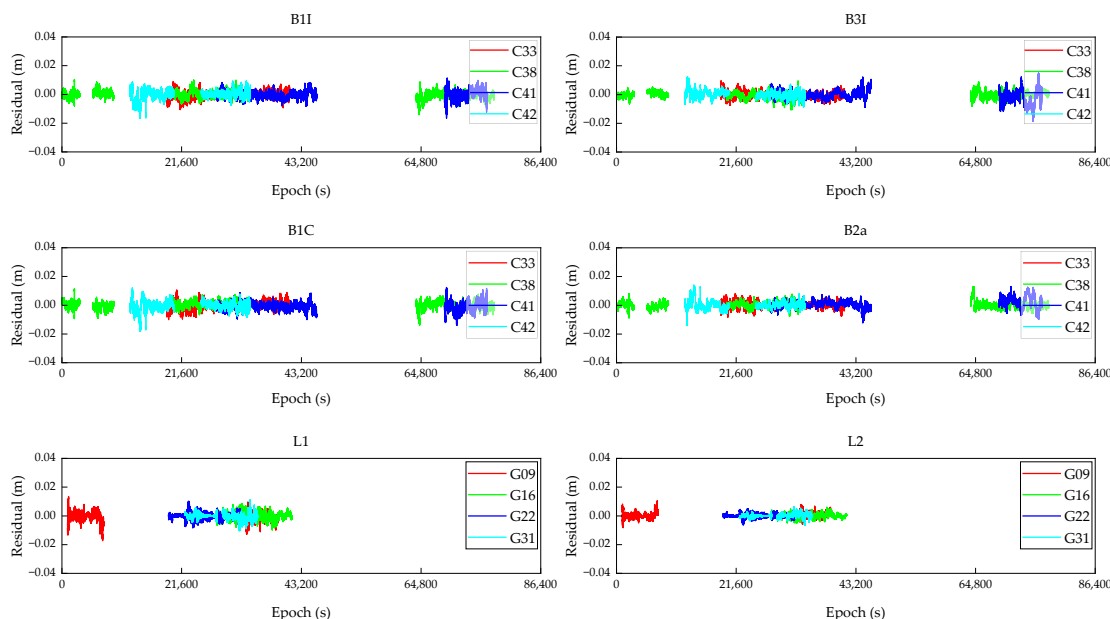

**Figure 17.** The residual values corresponding to different frequencies of satellites for rover station $U_2$.

## 4. Discussion

By making full use of the BDS-3 multi-frequency data, this paper studied the multi-frequency long-range URTK method of the BDS-3 in detail, using the most mature global navigation satellite system GPS as a comparison object. The proposed method is simple and easy to implement and can achieve centimeter-level positioning without introducing other high-precision products. Also, it is suitable for the BDS-3, GPS, and Galileo systems, which use code division multiple access technology for signal modulation. In addition, it lays a foundation for the next multi-system integration.

In this study, before analyzing the experimental results, a simple analysis of the selected two sets of long-range reference station observation data was performed. Since the data used in this study were collected in China, the satellite visibility of the GPS was slightly lower than that of the BDS-3, which could be the main reason for the poor performance of the GPS. Then, the results of the most critical parts of the three modules were analyzed, and the analysis showed high consistency in the results. With the increase in the frequency of the BDS-3 observation data, the success rates of the AR, convergence time, stability, and positioning accuracy were significantly improved, which fully reflects the advantages of the BDS-3 multi-frequency data. Further, the distance between the reference stations of reference station network 1 was smaller than that of reference station network 2, so the result of reference station network 1 was better than that of reference station network 2 in the analysis process.

Currently, the GPS has been in the stage of modernization, and the Galileo system has been constantly improving. In addition, multi-frequency observation data have become a mainstream trend. However, the issue of how to use the massive data rationally has been a problem requiring further research and analysis. The data selected in this study were collected in China, so they cannot fully represent the global positioning performance of the BDS-3, and global verification needs to be conducted in future work. At the same time, the observation region selected in this study was relatively simple, and more complex environments need to be studied in the future, which requires the effective use of multi-system data.

Therefore, in future work, the following three aspects could be addressed:

1. Realization of a multi-system multi-frequency long-range undifferenced network RTK method;
2. Verification of the global performance of the BDS-3, using the observation data of different regions in the world;
3. Studying positioning performance in a complex environment.

## 5. Conclusions

This paper proposes a BDS-3 multi-frequency URTK method for a long-range reference station network. First, the double-difference multi-frequency phase integer AR model of a reference station network is constructed considering the atmospheric error. The double-difference integer ambiguity sum of the closed reference stations of zero is set as a constraint condition to realize accurate determination of the double-difference integer ambiguity of a reference station. Then, the double-difference integer ambiguity is converted into undifferenced integer ambiguity using a linear relationship of double-difference integer ambiguity between reference stations, and it is substituted into the observation equation of a reference station in order to calculate the comprehensive undifferenced error correction value. Considering different characteristics of the observation error, dispersion error (ionospheric delay error), and non-dispersion error based on the tropospheric delay error are calculated. The undifferenced error correction value of a rover station is calculated using inverse distance weighted interpolation; finally, the error correction and high-precision positioning of the rover station are realized.

Using the CORS network data, the positioning process and accuracy analysis of B1I/B1C/B2a/B3I, B1C/B2a/B3I, B1I/B2a/B3I, B1C/B2a, B1I/B3I, and GPS are performed. The experimental results show that the B1I/B1C/B2a/B3I, B1C/B2a/B3I, and B1I/B2a/B3I are better than B1C/B2a, B1I/B3I, and GPS in terms of their AR success rate, stability, convergence time, and positioning accuracy; also, the positioning performance of the BDS-3 is generally better than that of the GPS system. Comparing the positioning results of each combined observation with the known values of coordinates, the deviations in the E, N, and U directions of coordinate components are calculated to be at the centimeter level; thus, centimeter-level positioning can be achieved using the proposed method. The proposed method has good application prospects in various fields, including agricultural production, location service, and deformation monitoring.

The results have indicated that multi-frequency data can significantly improve the positioning performance of the URTK. However, in future work, it is necessary to study long-range URTK positioning performance for the gathering of further multi-system multi-frequency observation data.

**Author Contributions:** H.Z. conceived the idea and designed the experiments with J.Z., J.L., A.X. and H.Z. and wrote the main manuscript. J.Z., J.L. and A.X. reviewed the paper. All components of this research were carried out under the supervision of H.Z. and A.X. All authors have read and agreed to the published version of the manuscript.

**Funding:** This research was funded by the National Natural Science Foundation of China (Nos. 42030109, 42074012) and the open fund of the State Key Laboratory of Satellite Navigation System and Equipment Technology (No. CEPNT-2018KF-13), and it was supported by the LiaoNing Revitalization Talents Program (Nos. XLYC2002101, XLYC2008034, XLYC2002098).

**Data Availability Statement:** The authors gratefully acknowledge Kepler for providing the multi-GNSS data. The other datasets are available from the corresponding author on reasonable request.

**Acknowledgments:** The data in this paper were provided and allowed to be used by Kepler Satellite Technology Co Ltd., Wuhan, China. The authors gratefully acknowledge Kepler for providing the multi-GNSS data.

**Conflicts of Interest:** The authors declare no conflict of interest.

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
