# Peer review of "An Efficient BDS-3 Long-Range Undifferenced Network RTK Positioning Algorithm"

_remotesensing, doi:10.3390/rs15164060_

Round 1

Reviewer 1 Report

This paper proposes a new method for achieving long-range undifferenced network RTK positioning in the BeiDou multi-frequency scenario. The method utilizes undifferenced ambiguity resolution to extract undifferenced augmentation corrections. Then the undifferenced corrections are separated into dispersion corrections and non-dispersion corrections, enabling differentiated interpolation based on different spatial correlations. The approach is innovative and provides practical guidance for the application of long-range network RTK positioning. However, I still have some questions regarding the experimental design and results section, which are listed as follows:

1. The paper implements a long-range undifferenced network RTK algorithm based on multi-frequency data. This method projects the fixed double-difference ambiguity into undifferenced ambiguity. The atmospheric delay is calculated by undifferenced observations to realize atmospheric delay recovery and modeling. The rover station uses inter-satellite single-difference observations to achieve high-precision positioning. The principle of this method is similar to PPP-RTK on the mobile terminal. Please explain the advantages and differences of this method compared with multi-frequency PPP-RTK.

2. The 2.2 section of the paper introduces the classification error interpolation method in detail, and obtains the user 's undifferenced error correction values from the interpolation coefficient. Although the classification process is described in detail, the interpolation coefficients are given without the calculation of the interpolation coefficients, so please provide additional information in the corresponding section.

3. Equation (5) gives the random walk equation of ionospheric delay error and tropospheric delay error, and gives the mean and variance of the inter-epoch variation in the form of normal distribution. As a key parameter in random walk, how power spectral density is related to variance. In this paper, the power spectral density values used in line 224-225 industries are given, but how to obtain the variance is not explained in detail. It is suggested that the relationship between power spectral density and variance should be described more deeply.

4. The keyword "long-range network real-time kinematic" is too long and it is suggested to divide it into: "long-range", network RTK".

5. Equation (5), Eq. (6), equation (13). Please Unify your description.

6. This paper is based on a undifferenced algorithm, but the derivation of the formulas still performs differencing, could the authors please explain how the undifferenced are reflected in this paper?

7. The description of undifferenced should be unified in the text, e.g. "undifferenced" in line 264 and "non-difference" in line 265, should be unified in the text.

8. The description of frequencies in line 244 of the text uses B1, B2, and B3 signals, and it is recommended that the type of signals be described more specifically, such as in the form of B1C/B2a and B1I/B3I in section 3.

9. In Table 8, the positioning result of u1 in U direction using B1l/B3l is better than it using other frequencies. A comment should be provided for this result.

The English language of the manuscript needs polishing.

Reviewer 2 Report

REVISION MANUSCRIPT Remote Sensing- 2534702: An Efficent BDS-3 Long-range Undifferenced Network RTK Positioning Algorithm.

General comments:

The authors proposed a long-range undifferenced network RTK (URTK) algorithm based on multi-frequency observation data of the BeiDou-3 global navigation satellite system (BDS-3). To accomplish that, the authors first designed the multi-frequency phase integer ambiguity resolution (AR) model and in a second step, it was determined. Therefore, I found the research interesting. However, the kinematic positioning capabilities have been researched and published several times. I believe some minor suggestions and revisions need to be applied in order to consider it for publication.

Specific comments:

Comment 1: The Introduction section is too long. The authors need to be more specific about relevant scientific literature regarding their study.

Comment 2: What is the precision or accuracy of the reference station’s coordinates? What is the reference system you are using for your RTK solution?

Comment 3: Line 333 to 335, I am not quite sure if in the long-range RTK approach is possible to agree with your statement: “tropospheric delay error, ionospheric delay error, are completely eliminated or significantly weakened”.

Comment 4: What is the reason of selecting 1 s for the frequency interval and a 15-deg. elevation angle?

Comment 5: Figure 3: any chance to avoid reference stations located on top of buildings?

Comment 6: Figure 4: I believe this can be improved. In fact, what this figure shows is the location of the stations (I do not see the values for the azimuths).

Comment 7: Figure 5: I see the three (top) figures very identical (no difference among them). Also, you need to name them as (a), (b), etc.

none
